# Dietary intake, body composition and micronutrient profile of patients on maintenance hemodialysis attending Kiruddu National Referral Hospital, Uganda: A cross sectional study

Fred Lawrence Sembajwe[1], Agnes Namaganda[1,2]*, Joshua Nfambi[1], Haruna Muwonge[1], Godfrey Katamba[3], Ritah Nakato[4], Prossy Nabachenje[5], Enid Kawala Kagoya[6], Annet Namubamba[7], Daniel Kiggundu[8], Brian Bitek[9], Robert Kalyesubula[1], Jehu Iputo[2]

1 Department of Medical Physiology, Faculty of Health Sciences, Busitema University, Mbale, Uganda, 2 Department of Medical Physiology, College of Health Sciences, Makerere University, Kampala, Uganda, 3 Department of Physiology, College of Health, Medicine and Life Sciences, King Ceasor University, Kampala, Uganda, 4 Department of Pharmacology, College of Medicine, Lira University, Lira, Uganda, 5 Department of Pediatrics, Faculty of Health Sciences, Busitema University, Mbale, Uganda, 6 Department of Community Health and Behavioural Sciences, Faculty of Health Sciences, Busitema University, Mbale, Uganda, 7 Department of Public Health, School of public Health Kololo Annex, Makerere University, Kampala, Uganda, 8 Department of Medicine, Dialysis Unit, Kiruddu National Referral Hospital, Kampala, Uganda, 9 Department of Pharmacology and Therapeutics, Faculty of Medicine, Gulu University, Gulu, Uganda

* namutoagnes@gmail.com

## Abstract

Patients on maintenance hemodialysis are at a great risk for altered nutritional status, characterized by protein energy wasting and micronutrient deficiency due to medication interactions and dietary restrictions. This study determined the dietary intake, micronutrient profile and body composition of patients on maintenance hemodialysis at Kiruddu National referral hospital (KNRH), Uganda. A cross sectional study was done among adult CKD patients on maintenance hemodialysis therapy at KNRH. Data concerning patients' demographics, clinical history and dietary intake was obtained using interactive and quantitative food frequency questionnaires. Body composition was obtained using the TANITA BC-351, Japan weighing Bathroom scale and anthropometric measurements using standard methods and procedures. Serum micronutrient profile assessment was done using the COBAS Auto analyzer. Data analysis was done using the SPSS software version 20. T-test was used to make comparisons and logistic regression analysis was done to check for any correlations. A P-value of < 0.05 was considered statistically significant. Among the 63 hemodialysis patients, 38% were female, with a median duration of hemodialysis of 12 months and the overall age range of patients was 31–40 years. Majority (92.1%) of the patients had hypertension. Carbohydrates like maize flour were highly consumed, in addition to eggs among the proteins on the daily basis. Fruits and vegetables were not highly consumed. Regarding body composition; 75% of the study participants had normal Body mass Index (BMI), the mean muscle mass was 51.94±8.68, body fat was 15.25±7.35, bone mass was 2.77±0.46 and body

**Data Availability Statement:** All relevant data are within the manuscript and its Supporting Information files.

**Funding:** We received funding from the Fogarty International Center of the National Institutes of Health (NIH), U.S. Department of State's Office of the U.S. Global AIDS Coordinator and Health Diplomacy (S/GAC), and President's Emergency Plan for AIDS Relief (PEPFAR) under award number: IR25TWO11213. The funders had no role in study design, data collection and analysis, decision to publish, or preparation of the manuscript.

**Competing interests:** The authors have declared that no competing interests exist.

water was 62.04±9.06. Patients had deranged micronutrient levels especially for Vitamin D, Potassium and phosphorus. In conclusion, hemodialysis patients at KNRH, have altered nutritional status as evidenced by altered body weight for some patients and deranged micronutrient levels. We recommend that hemodialysis patients should be regularly assessed for nutritional status, appropriately treated and educated about their nutritional status.

## Introduction

Chronic kidney disease (CKD) is a public health problem globally, that affects approximately 8–16% of the world's population [1, 2]. It is recognized as a non-communicable disease (NCD) with high morbidity and mortality in Sub-Sahara Africa (SSA) [3]. The prevalence of CKD in Uganda varies by region whereby: in eastern Uganda it is 12.5%, in southwestern Uganda is 3.9% [1], estimated to be about 14.4% in the northern region [4] and in Central Uganda is 2.5% [3].

In CKD, the kidney functions progressively decline, leading to changes in requirements and utilization of various nutrients which increases the risk of malnutrition in this population [5]. A previous study related to the nutritional status has shown that malnutrition or wasting is common in CKD, as approximately 18–75% of patients with CKD, undergoing maintenance dialysis therapy, showed evidence of wasting [6]. A study conducted in Uganda, at Mulago National Referral Hospital (MNRH) found that 47.3% of CKD participants attending the renal clinic had Protein Energy Wasting [7].

CKD patients experience uremic symptoms and have inflammation which can contribute to poor dietary intake and an increase in metabolic stress. Collectively, this leads to a reduction in body protein synthesis, hence a decline in nutritional status, termed as malnutrition [8]. The American Society for Parenteral and Enteral Nutrition has defined malnutrition as "an imbalance of need and supply of energy, protein and micronutrients leading to growth and development defect" [9]. Dietary intake is a major determinant of over or under nutrition, but it is not the only influence on an individual's nutritional status [10]. Restrictions imposed on CKD patients aiming at reducing protein, phosphate, or potassium intake make it difficult to ensure adequate micronutrient content in the diet [11]. Bross et al, suggests that examining the quality and quantity of diverse types of nutrients is also critical in assessing the dietary intake in these individuals since they need to extensively modify their nutrient intake [12]. Thus, it is imperative to do a dietary evaluation for CKD patients.

CKD patients are at great risk for micronutrient deficiency, since they are put on dietary restrictions and have reduced appetite which may contribute to comorbidities such as anemia, cardiovascular disease, and metabolic imbalances [13]. Micronutrient malnutrition affects a third to half of the global population [14]. Several micronutrients are important in CKD patients including dietary vitamins (thiamin, riboflavin, niacin, pyridoxine, folate, cobalamin, vitamins A, D, E, and K) and minerals (sodium, potassium, calcium, magnesium, phosphorus, selenium, and zinc) [15]. The most affected micronutrients at all stages of CKD include; Ascorbic acid, Thiamine, Pyridoxine, folic acid, zinc and selenium [11]. Iorember et al, reported that CKD patients including dialysis and non-dialysis patients show a decrease in the intake of micronutrients such as vitamins, folate, iron, and pantothenic acid [13]. Thus, better understanding of the micronutrient availability in CKD patients could have an impact on many complications linked to vitamin and trace element disorders, including high mortality,

increased risk of atherosclerosis, inflammation, oxidative stress, anemia, polyneuropathy, encephalopathy, weakness and fragility, muscle cramps, bone disease, depression, or insomnia that could occur in CKD.

Body composition reflects nutritional intakes, losses and needs over time. It may also assist in identifying physical dysfunction earlier in CKD patients. Body composition is frequently altered among patients with CKD, whereby obesity and muscle wasting are common, sometimes occurring simultaneously [16]. It has been previously suggested that body composition determination is an important predictor of survival in chronic hemodialysis patients [17–19]. Therefore, assessment of body composition will provide an in-depth understanding of weight changes in terms of describing obesity and/or muscle wasting that could allow for earlier and objective management of under nutrition in CKD patients. The progressive loss of kidney function observed among CKD patients results in increased risk of malnutrition [20]. Malnutrition in these patients reduces quality of life, increases the risk of infection, increases the risk of diseases, and impairs wound healing. It also results in poor rehabilitation, fatigue, lethargy, as well as increased hospitalization and mortality in these patients [9]. This study, therefore, was aimed at establishing the dietary intake, micronutrient profile and body composition of CKD patients at Kiruddu National Referral Hospital in Central Uganda.

## Methods and materials

### Study design

We conducted a cross sectional study involving CKD patients on maintenance hemodialysis, aged 18 years and above.

### Study setting

The study was conducted at Kiruddu National Referral Hospital (KNRH). KNRH is one of the National referral hospitals in the Uganda. The care system is public and it also serves as the teaching hospital of Makerere University, College of Health Sciences. The hospital offers inpatient and outpatient services. It has an official bed capacity of 200 beds. The outpatient services offered include; renal, diabetic, hypertension, respiratory, etc. to 500 patients on a daily basis (KNRH records, Dec 2020). The hospital receives on average five (new) and 10 (old) outpatient CKD clients on the renal clinic day and over 100 CKD patients on the dialysis unit (KNRH records, Nov 2020).

### Sampling of study participants

Convenience sampling procedure was used to enroll patients as they were registered on the renal unit in-patient and out-patients' register. The participants were met before being connected to the dialysis machine.

Patients who were already diagnosed with chronic kidney disease and attending the dialysis unit for more than 2 months were recruited. Consent was sought first from the participants, after adequate information about the study objectives, procedure and benefits of the study was given. The participants were then asked to sign on the consent forms. This was separated from the questionnaire. The questionnaires were coded by labeling with numbers from 001 to 063 to ensure confidentiality.

### Inclusion criteria

CKD patient on maintenance dialysis aged 18 years and above.

## Exclusion criteria

CKD patients on peritoneal dialysis, with Nephrotic syndrome, recent peritonitis (less than 4 months), malignancy and infectious diseases (tuberculosis), and those found to be on steroid therapy.

## Sample size estimation

The KNRH medical records office showed that the hospital receives 120 CKD patients monthly (KNRH records, 2022). We used the Yamane formula developed by Taro Yamane in 1967 with a margin of error 5% to calculate the sample size of 92 participants. We enrolled 92 participants, however 63 were eligible.

## Study instruments

A structured pretested questionnaire was used to capture several socio-demographic characteristics (age, sex, marital status, highest level of education and occupation), lifestyle habits and family history (current smoking status, physical activity, current alcohol use, family history of diabetes, CKD and obesity, and use of concomitant medication) and several clinical measurements including weight, height, and blood pressure.

## Study procedures

**Dietary intake assessment.** Trained fieldworkers administered interactive Semi-quantitative food frequency questionnaires to the study participants as described by [12]. The Food frequency questionnaire (FFQ) used in this study was adapted from the WHO FFQ and adapted for this study. It was checked for content, construct and internal validity. It was also tested and retested for consistency and inter rater reliability.

**Anthropometric measurement.** including: body weight, height and waist, hip and upper arm circumference were measured to the nearest 0.1Kg or 0.1cm respectively, using a portable weight scale (for weight); stadiometer for height; waist and mid-upper arm circumferences were obtained using a flexible measuring tape. Mid- Upper Arm circumference (MUAC) was obtained from the non-dominant hand using the adult MUAC scale (or tape measure).

**Body composition assessment.** was done using a (TANITA BC-351, Japan) weighing bathroom scale. It uses the principle of different impedances of the conductive and insulated parts of the body, calculates the weight and proportion of various components in the body, automatically analyzes the test data, and uses the chart method to visually illustrate physical health [21]. The scale was switched on, then participant's age, sex and height were entered. The participant was then asked to step on the scale and the readings for body mass index, muscle mass, abdominal fat level, basal metabolic rate, bone mass, metabolic age, total fat percentage and total body water were automatically generated.

**Sample collection and storage.** Blood was collected in a serum separator vacutainer and centrifuged to obtain serum. Serum was stored at 4–8°C in the hospital for a few hours, until shipment took place later in the afternoon using ice boxes (with frozen ice packs) to Mulago National Referral Hospital-Clinical chemistry laboratory, for further analysis.

## Micronutrient profile assessment

Serum micronutrient levels were assessed using the COBAS INTEGRA 400 PLUS chemistry analyzer (Roche Diagnostics Ltd. Rotkreuz Switzerland). Blood samples were taken from CKD patients before beginning their dialysis session for the assessment of the biochemical parameters Vitamin D, sodium, potassium, calcium, magnesium, and phosphorus to obtain the serum

micronutrient levels. All investigations were carried out in the Clinical Chemistry laboratory of Mulago National Referral hospital.

### Data collection period

Participants were recruited into the study during the period of January 2022 to February 2022.

### Ethical clearance

Approval for the study was obtained from the Research Ethical Committee (REC) of Mbale regional referral Hospital (reference Number MRRH 2021–57). Administrative clearance was obtained from the administration of Kiruddu National Referral Hospital. Written informed consent was obtained from the participants before obtaining any samples or data from them. Confidentiality was also maintained during data collection by coding or labelling the questionnaire with a serial number instead of participants' names.

### Data analysis

Data obtained was entered in Microsoft excel sheet, then exported to the SPSS software, version 20 for analysis. Descriptive statistics such as means and ranges of variables of interest were computed. T-test was used to make comparisons and logistic regression analysis was done to check for any possible correlations in our data. P value < 0.05 was considered statistically significant.

## Results

We summarized the socio-demographic characteristics of our study participants (n = 63) in Table 1, where majority of them were males at about 62% and mostly aged between 31 to 40years.

Table 1 also shows that 52.4% of our participants were married or cohabiting and (34.9%) had attained secondary education. The unemployed participants were 36.5%, 23.8% had 4 children and 28.6% of the respondents were Pentecostals. We also obtained medical history of our study participants as shown in Table 2, whereby 7.9% of the Patients had Diabetes Mellitus 63 (5) with 40% being newly diagnosed.

In addition, 92.1% had Hypertension (HTN) 63(58), whereby 46.6% of them had been living with the condition for over 3 years (Table 2).

### Dietary intake among the participants

The commonly eaten carbohydrate is maize flour whereby 29% consumed it on a daily basis, followed by rice 27%, cassava 15%, soya flour 12%, sweet potatoes 5%, cassava flour 5%, millet flour 2%, matooke 2%, Irish potatoes 2%, macaroni/spaghetti (pasta) 1% and yams 0% (Fig 1).

Among protein sources; Eggs were highly consumed, whereby 22% ate it on a daily basis, followed by milk (21%), beans (19%), ground nuts (9%), and least eaten were a variety meats/offal ("byenda") (1%) (Fig 2).

Fruits and vegetables were not highly consumed as only 9% consumed apples, 2% consumed oranges, mangoes and 1% consumed passion fruits, jackfruit and pineapples (Fig 3).

Among vegetables; 3% consumed cabbage, 2% avocado, 1% greens and then pumpkin was never eaten (Fig 3).

**Table 1. Social demographic characteristics of the study participants.**

| Variable | Frequency n = 63 | Percentage (%) |
|---|---|---|
| Age category | | |
| 18–30 | 12 | 19.0 |
| 31–40 | 21 | 33.3 |
| 41–50 | 19 | 30.2 |
| 51–60 | 7 | 11.1 |
| 61–70 | 2 | 3.2 |
| 71–80 | 2 | 3.2 |
| Gender | | |
| Female | 24 | 38.1 |
| Male | 39 | 61.9 |
| Religious affiliation | | |
| Anglican | 14 | 22.2 |
| Catholic | 13 | 20.6 |
| Islam | 14 | 22.2 |
| Pentecostal | 18 | 28.6 |
| SDA | 4 | 6.4 |
| Highest level of education attained | | |
| None | 1 | 1.6 |
| INTERMEDIATE | 1 | 1.6 |
| Primary | 12 | 19.0 |
| Secondary | 22 | 34.9 |
| Tertiary (diploma certificate) | 9 | 14.3 |
| University | 18 | 28.6 |
| Marital status | | |
| Divorced/Separated | 8 | 12.7 |
| Married/Cohabiting | 33 | 52.4 |
| Single | 20 | 31.7 |
| Widowed | 2 | 3.2 |
| Number of Children | | |
| None | 14 | 22.2 |
| One | 8 | 12.7 |
| Two | 7 | 11.1 |
| Three | 7 | 11.1 |
| Four | 15 | 23.8 |
| Five and above | 12 | 19.0 |
| Occupation | | |
| Farmer | 4 | 6.3 |
| Formal employment | 13 | 20.6 |
| Informal employment | 5 | 7.9 |
| PASTOR | 1 | 1.6 |
| STUDENT | 2 | 3.2 |
| Self-employed | 11 | 17.5 |
| TECHNICIAN | 1 | 1.6 |
| Unemployed | 26 | 41.3 |

**Table 2. Medical history of the study participants.**

| Variable | Frequency n = 63 | Percentage (%) |
|---|---|---|
| Diabetes Mellitus (DM) | 5 | 7.9 |
| Diabetes Mellitus Duration | | |
| 1-2yrs | 1 | 20.0 |
| 3yrs and above | 2 | 40.0 |
| Newly diagnosed | 2 | 40.0 |
| Hypertension (HTN) | 58 | 92.1 |
| Hypertension Duration | | |
| 1-2yrs | 17 | 29.3 |
| 3yrs and above | 27 | 46.6 |
| Newly diagnosed | 7 | 12.1 |
| Missing | 7 | 12.1 |

## Body composition analysis findings were

Mean Body water composition for all study participants was 62.040±9.06, body fat 15.248 ±7.353, Muscle mass 15.942±8.68 and bone mass (2.770±0.46). The Anthropometric measurement findings were; mean BMI was 23.45±5.378 with 75% 48(61) normal BMI, 2% 1(61) underweight, 16% 10(61) overweight and 7% 4(61) obese. The mean MUAC was 25.952±3.679 cm, Waist circumference was 85.533±12.443 cm, Hip circumference was 92.746±14.238 cm and Waist Hip ratio (WHR) was 15.248±7.353 (Tables 3 and 4).

## Micronutrient level analysis findings were

The mean Vitamin D levels were 33.883±15.184 (ng/ml) where 27% 14 (60 participants) had normal levels and 73% 46 (60 participants) had low levels of vitamin D. Mean Albumin levels were 31.883±4.786 where 70% 46(60 participants) had reduced levels and 30% 16(60

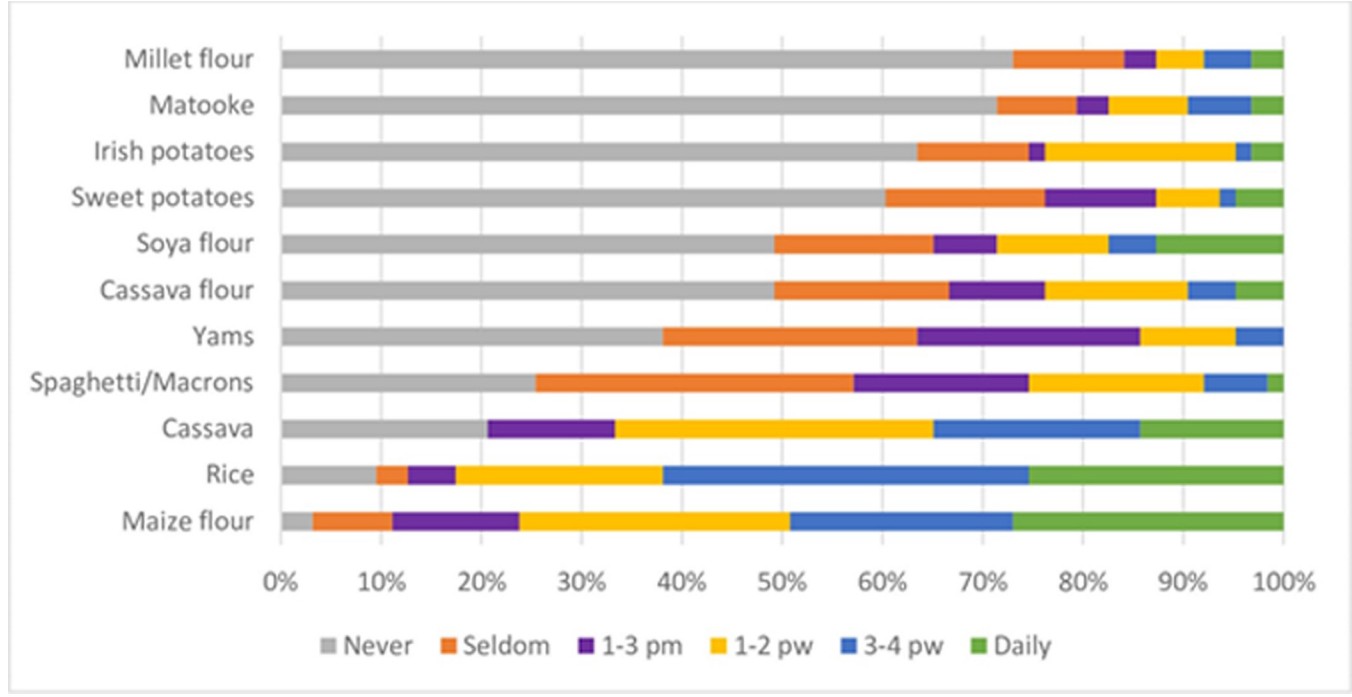

**Fig 1. Frequency of dietary intake of carbohydrates.** pm = per month; pw = per week.

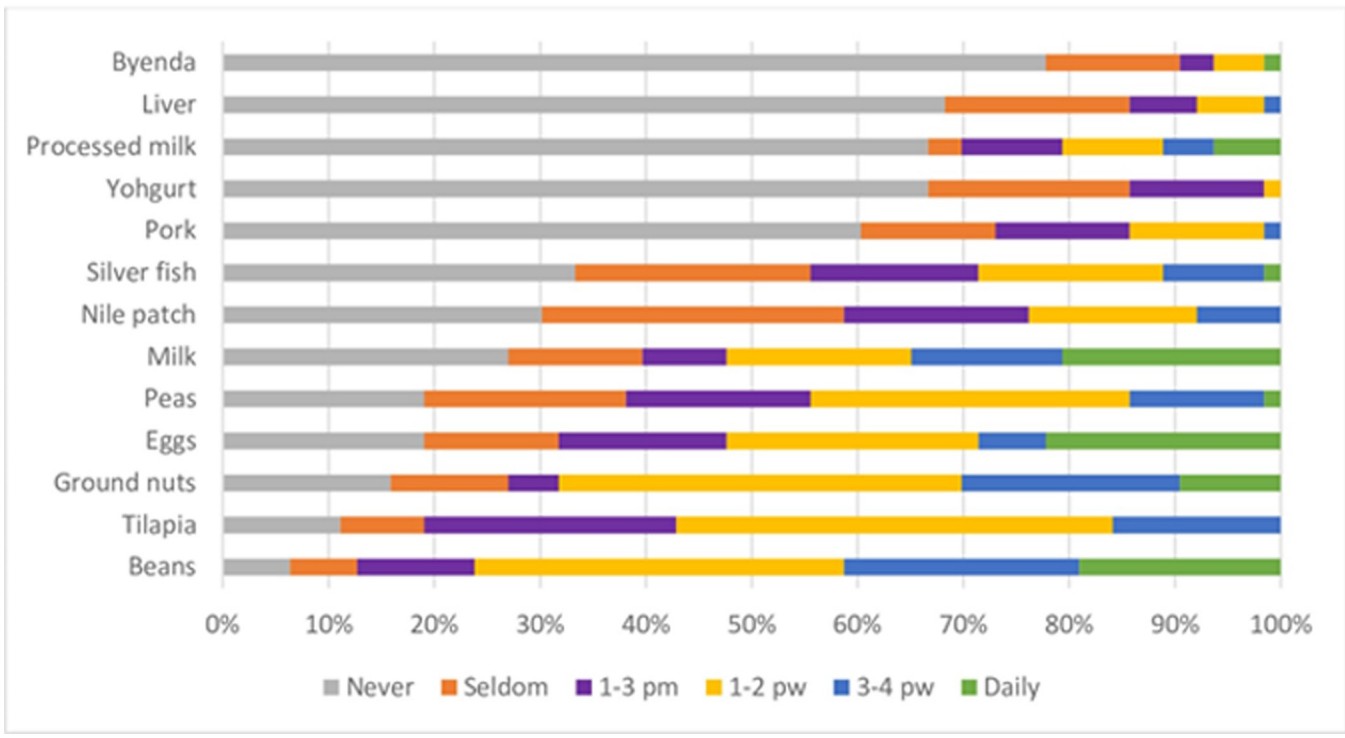

**Fig 2. Frequency of dietary intake of proteins.** pm = per month; pw = per week.

participants) had normal levels. Mean sodium levels were 135.917±8.363 mEq/L whereby 25% 13(60 participants) had raised levels and 75% 47(60 participants) had normal levels. Mean Potassium levels were 5.761±1.171 mEq/L whereby 56% 36(60 participants) had raised levels and 44% 24(60 participants) had normal levels. Mean Chloride levels were 101.489±6.626 mEq/L, whereby 17% 11(60 participants) had abnormal levels. Mean Calcium levels were 2.177±0.35 mEq/L, whereby 57% 36(60 participants) had raised levels and 43% 24(60 participants) had normal levels. Lastly, mean phosphorus levels were 1.73±0.72 mEq/L, whereby 48% 27(60 participants) had raised levels and 52% 33(60 participants) had normal levels (Tables 5 and 6).

## Malnutrition

Was present in 27% of the patients of which 64.7% were males, however; 73% were well nourished as determined by the BMI (Table 7).

Additionally, among the malnourished patients, majority had unhealthy WHR, HTN and were males. The level of education attained was significantly (p = 0.045) associated with increased risk for malnutrition following a multivariate logistic regression analysis (Table 7). In Table 8, body weight significantly increased with increase in total body fat percentage, bone mass, body water and muscle mass percentage, while, waist circumference only significantly increases with increase in abdominal fat level (p<0.05) (Table 9).

## Discussion

### Dietary intake

Adequate nutrition is necessary for ensuring recovery or survival among patients suffering from various illnesses. Dietary assessment is of paramount importance in providing optimal

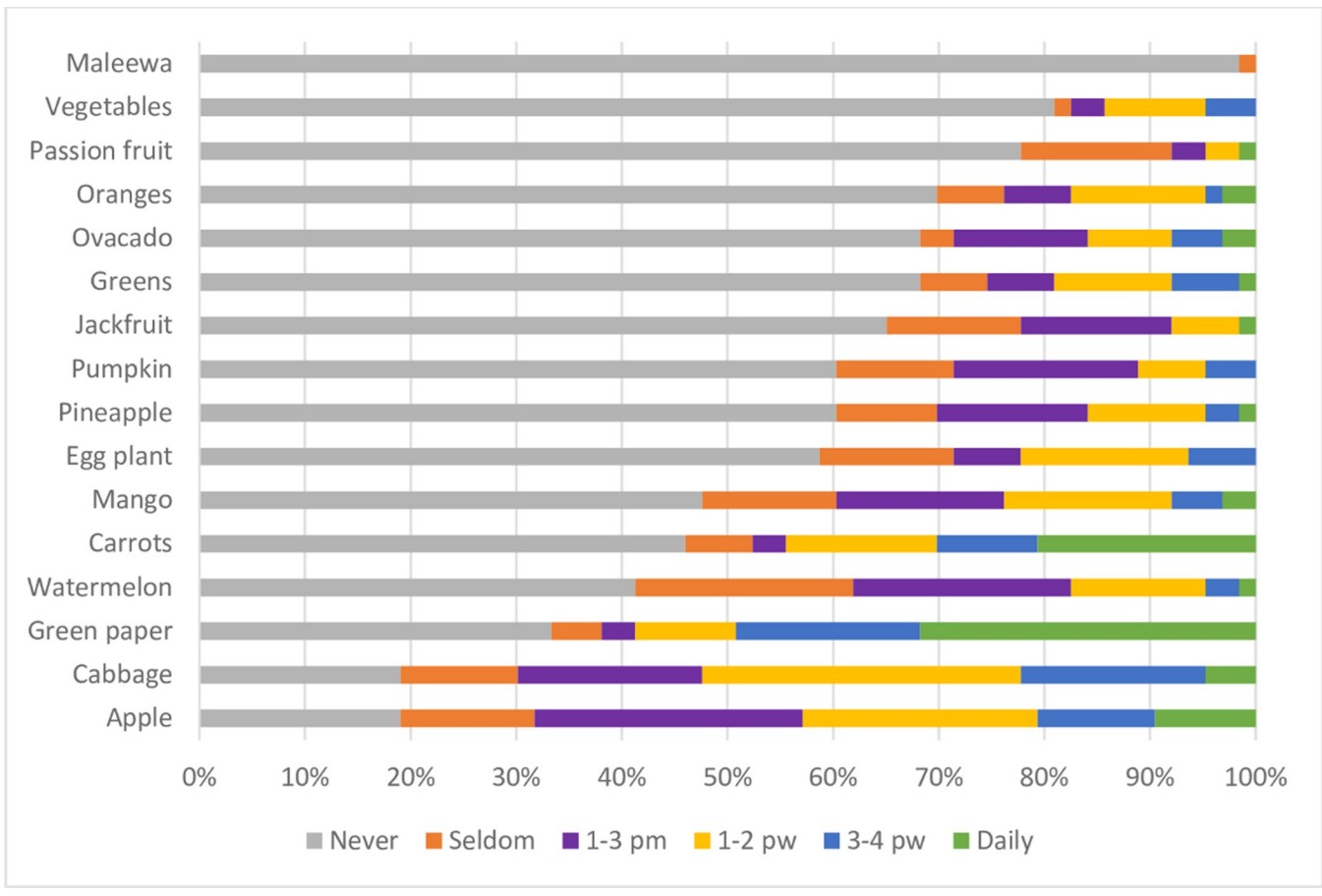

**Fig 3. Frequency of dietary intake of fruits and vegetables.** pm = per month; pw = per week.

**Table 3. Clinical observations and assessment of body composition.** Descriptive Statistics (n = 63).

| Variable | Mean±Std.Dev | Min | Max |
|---|---|---|---|
| **Clinical Observations** | | | |
| Systolic blood pressure (mmHg) | 150.887±26.52 | 85 | 230 |
| Diastolic blood pressure (mmHg) | 92.081±21.17 | 55 | 165 |
| Height(cm) | 168.503±7.855 | 152.2 | 187.2 |
| Weight(cm) | 64.979±12.294 | 43.9 | 101.1 |
| **Assessment of body composition** | | | |
| Body mass index (BMI) | 23.47±5.378 | 17.6 | 55.1 |
| MUAC (cm) | 25.952±3.679 | 19.3 | 37.2 |
| Abdominal fat | 4.387±3.296 | 1 | 14 |
| Waist circumference (cm) | 85.533±12.443 | 60 | 132.08 |
| Hip circumference (cm) | 92.746±14.238 | 48.4 | 121.92 |
| Waist Hip Ratio (WHR) | 0.941±0.201 | 0.701 | 2.012 |
| Total body fat | 15.248±7.353 | 5 | 38.2 |
| Muscle mass | 51.942±8.68 | 34.9 | 72.8 |
| Bone mass | 2.770±0.46 | 1.9 | 3.9 |
| Body water (L) | 62.040±9.06 | 42.8 | 83.2 |

**N.B.:** MUAC = middle upper arm circumference; BMR = basal metabolic rate; cm = centimeters; mmHG = millimeters of Mercury; L = liters

**Table 4. Assessment of body composition.**

| Variable | Frequency(n = 63) | Percent |
|---|---|---|
| Body Mass Index (BMI) | | |
| Normal | 46 | 75.41 |
| Underweight | 1 | 1.64 |
| Overweight | 10 | 16.39 |
| Obese | 4 | 6.56 |
| Missing data | 2 | |
| Abdominal fat level) | | |
| Healthy | 59 | 98.39 |
| Excessive | 1 | 1.61 |
| Missing data | 3 | |
| Waist circumference (cm) | | |
| Healthy WC | 25 | 39.68 |
| Moderately unhealthy WC | 13 | 20.63 |
| Severely unhealthy WC | 25 | 39.68 |
| Waist hip ratio | | |
| Healthy WHR | 18 | 28.57 |
| Unhealthy WHR | 45 | 71.43 |

N.B.: BMI = body mass index; WC = waist circumference; WHR = waist hip ratio; cm = centimeters

care to individuals with CKD, and in particular to dialysis patients, even though it is not performed routinely in these patients.

Our study findings showed that among Carbohydrates consumed by the participants, cassava, maize flour and rice were the most frequently consumed foods, which is not as surprising as these are some of the major staple foods in Uganda (Fig 1). Due to the high potassium content of some foods such as plantain bananas (matooke) and potatoes, it is easy to understand why very few patients regularly consumed these foods, as high dietary potassium is detrimental to CKD patients. Indeed, previous studies have focused on devising the best method of regulating or reducing potassium levels in the bananas (matooke), potatoes and vegetables consumed by CKD patients [22, 23].

Among Proteins consumed by the participants, beans were the most consumed source of proteins as compared to the other sources particularly the animal proteins (Fig 2), which

**Table 5. Summary descriptive statistics for micro nutrient levels.**

| Variable | Obs | Mean | Std.Dev. | Min | Max |
|---|---|---|---|---|---|
| **Micro nutrient** | | | | | |
| vitD (ng/ml) | 63 | 33.883 | 15.184 | 10.2 | 114.1 |
| Alb (g/dL) | 63 | 31.883 | 4.786 | 13 | 39 |
| Na (mEq/L) | 63 | 135.917 | 8.363 | 97 | 157 |
| K (mEq/L) | 63 | 5.761 | 1.171 | 3.62 | 9.15 |
| Cl (mEq/L) | 63 | 101.489 | 6.626 | 77.4 | 115.7 |
| $Ca^{2+}$ (mEq/L) | 63 | 2.177 | 0.35 | 1.52 | 3.08 |
| Phosp (mEq/L) | 63 | 1.73 | 0.72 | 0.74 | 4.37 |

N.B: VitD = vitamin D; alb = albumin; Na = sodium ions; K = potassium ions; Cl = chloride ions; $Ca^{2+}$ = calcium ions; Phosp = phosphorus; Obs = observations; Std Dev = standard deviation; ng/ml = nanograms/milliliters; g/dL = grams/liter; mEq/L = milliequivalents/liter

**Table 6. Micronutrient levels in the study participants.**

| Variable | Frequency | Percent |
|---|---|---|
| Vit D | | |
| Normal range | 14 | 26.98 |
| Out of range | 46 | 73.02 |
| ALB | | |
| Out of range | 44 | 69.84 |
| Normal range | 16 | 30.16 |
| Na | | |
| Out of range | 13 | 25.40 |
| Normal range | 47 | 74.60 |
| K | | |
| Out of range | 36 | 57.14 |
| Normal range | 24 | 42.86 |
| Cl | | |
| Out of range | 10 | 16.67 |
| Normal range | 50 | 83.33 |
| Ca2 | | |
| Out of range | 36 | 57.14 |
| Normal range | 24 | 42.86 |
| Phosp | | |
| Out of range | 27 | 47.62 |
| Normal range | 33 | 52.38 |

N.B.: 3 participants had missing data VitD = vitamin D; alb = albumin; Na = sodium ions; K = potassium ions; Cl = chloride ions; $Ca^{2+}$ = calcium ions; Phosp = phosphorus; Obs = observations; Std Dev = standard deviation

remain restricted among CKD patients [24, 25]. Our results show that most of our study participants were adhering to the health advice given by the clinical dietitians of restricting animal proteins and instead relying more on the plant sources of proteins such as beans [26]. But it might also be an indicator of the economic status of the majority of the participants that consumed beans as the most affordable source of protein compared to the other sources. For instance, going by the current market prices; the cost of purchasing one kilogram of beef can be used to buy three kilograms of beans, which would be consumed for almost a week unlike the beef that would be finished in one or two meals.

Our findings showed that among fruits consumed; apples are the commonly eaten fruits whereby 10% of the participants eat it on a daily basis, 10% eat it 3-4times, 26% eat it 1–3 times and 27% eat it 1–2 times (Fig 3). This implies that apples are readily available or affordable to most of the participants as they are sufficiently grown in the south western part of the country areas of Kabale and Kisoro districts. Then, cabbages and carrots remain the most consumed vegetables among our participants (Fig 3), which is also understandable, as these are among the cheapest and most common vegetables in most Ugandan fresh-food markets. Our study findings also showed that among the low nutrient density foods, chapatti is commonly eaten on a daily basis 18% and sausages are never eaten (Fig 4) since they are highly restricted among CKD patients.

The chapattis are quite conveniently prepared in form of 'rolex' (a local technique devised by many urban food vendors of rolling eggs and vegetables inside a chapatti for their customers) [27]. Alternatively many youths simply eat chapatti mixed with beans stew as a staple daily diet because it is readily available in most trading centers in Uganda [28].

**Table 7. Nutritional status (malnourished or well nourished) correlations.**

| Variable | All sample n = 63 | Nutritional status (BMI) n(%) | | AOR(95% CI) | P-value |
|---|---|---|---|---|---|
| | | Normal n = 46 | Malnourished n = 17 | | |
| **Gender** | | | | | 0.781 |
| Female | 24(38.1) | 18(39.1) | 6(35.3) | 1 | |
| Male | 39(61.9) | 28(60.9) | 11(64.7) | 0.2(0.01, 1.5) | 0.113 |
| **Highest level of education attained** | | | | | 0.045 |
| INTERMEDIATE | 1(1.6) | 1(2.2) | 0(0.0) | - | |
| None | 1(1.6) | 1(2.2) | 0(0.0) | 1 | |
| Primary | 12(19.0) | 11(23.9) | 1(5.9) | 0.04(0.02, 0.8) | 0.035 |
| Secondary | 22(34.9) | 19(41.3) | 3(17.6) | 0.05(0.03, 0.8) | 0.032 |
| Tertiary (diploma certificate) | 9(14.3) | 4(8.7) | 5(29.4) | 1.6(0.2, 11.2) | 0.639 |
| University | 18(28.6) | 10(21.7) | 8(47.1) | - | |
| Diabetes Mellitus (DM) | 15(23.8) | 10(21.7) | 5(29.4) | 3.2(0.6, 18.8) | 0.193 |
| Hypertension (HTN) | 58(92.1) | 42(91.3) | 16(94.1) | 1.4(0.1, 19.7) | 0.784 |
| Abdominal fat level | | | | | 0.087 |
| Healthy | 62(98.4) | 47(100.0) | 15(93.8) | - | |
| Excessive | 1(1.6) | 0(0.0) | 1(6.3) | - | |
| Waist hip ratio | | | | | 0.073 |
| Healthy WHR | 18(28.6) | 16(34.8) | 2(11.8) | 1 | |
| Unhealthy WHR | 45(71.4) | 30(65.2) | 15(88.2) | 4.2(0.5, 37.6) | 0.200 |

N.B.: HTN = hypertension; DM = diabetes mellitus; WHR = waist hip ratio; BMI = body mass index

## Micronutrient profile

Micronutrients are a critical component of a 'balanced diet' that is crucial not only in maintenance of a healthy body but also for facilitating the recovery process in patients of various ailments. Our findings show that the micronutrient levels for majority of the participants were out of range for vitamin-D, albumin, potassium and calcium (Tables 5 and 6). This might suggest the extent to which these micronutrients are restricted in the diet of our participants, but could also act as an indicator of the reduced ability of the damaged kidney to retain and preserve these particular micronutrients. Since the kidney is very important in the regulation of vitamin-D, potassium, calcium and retention of albumin in the human body, our results are indeed indicative of a severely damaged organ whose function is steadily declining. Since metabolic disorders such as hyperkalemia are closely associated with CKD [29, 30], dietary intake of additional potassium is highly restricted for this very reason. The hypokalemia observed in our study participants might be an indicator of prolonged use of potassium binding drugs in a bid to avoid the anticipated hyperkalemia [31–33]. The general micronutrient deficiency

**Table 8. Relationship between body weight, total body fat, body water muscle mass percentage and bone mass.**

| Body weight(kg) | Coef. | Std. Err. | t | P>t | [95% Conf. Interval] |
|---|---|---|---|---|---|
| Total body fat percentage | 1.00 | 0.11 | 8.85 | 0.001 | 0.78, 1.23 |
| Body water | 0.21 | 0.09 | 2.27 | 0.027 | 0.02, 0.40 |
| Muscle mass percentage | 0.81 | 0.14 | 5.61 | 0.001 | 0.52, 1.10 |
| Bone mass | 8.81 | 2.76 | 3.2 | 0.002 | 3.28, 14.33 |
| _cons | -29.33 | 7.41 | -3.96 | 0.001 | -44.19, -14.47 |

Prob > F = 0.0000, R-squared = 0.9298

**Table 9.  Relationship between waist circumference and total body water, abdominal fat level.**

| Waist circumference(cm) | Coef. | Std. Err. | t | P>t | [95% Conf. Interval] |
|---|---|---|---|---|---|
| Body water | 0.23 | 0.29 | 0.8 | 0.429 | -0.35, 0.82 |
| Total body fat percentage | 0.48 | 0.39 | 1.24 | 0.218 | -0.29, 1.26 |
| Abdominal fat level | 1.73 | 0.50 | 3.49 | 0.001 | 0.74, 2.73 |
| _cons | 56.29 | 22.78 | 2.47 | 0.016 | 10.70, 101.88 |

Prob > F = 0.0001, R-squared = 0.3164

among our study participants is not unusual and is consistent with what has been reported elsewhere, showing a correlation between declining kidney function and reduced plasma levels micronutrients such as vitamin D [34, 35]. The only surprise is that most of the study participants still had normal levels of sodium in their blood, which might imply that they were adhering to the medical advice of reducing dietary salt intake. But their out-of-range potassium levels are generally expected as has been reported elsewhere, with dyskalemia (out-of-range potassium levels) being associated with adverse outcomes in CKD [36–38].

## Body composition

Information about body composition is best analyzed using a comparison of the body anthropometric measurements or parameters. Indeed, the use of anthropometric parameter and biochemical measures as a means of determining nutritional status in CKD is also recommended in pediatrics as well [39]. These anthropometric parameters would include the BMI, MUAC, waist circumference, waist to hip ratio (WHR) and abdominal fat levels. In our study, the majority of the participants seem to have normal BMI and most of the other anthropometric parameters. However, due to the existence of outliers, with extremely low BMI and extremely high BMI, it is not appropriate to discuss the mean values, instead focus on the individual measures of interest. For instance, there was one individual with extremely low BMI of 17, which is an indication of being underweight and most probably malnourished. In addition, we also had an individual with an extremely high BMI of 55, which is abnormally high as a measure of being overweight, obese and malnourished in that regard. Indeed, we did not observe any significant correlation between nutritional status and body composition basing on

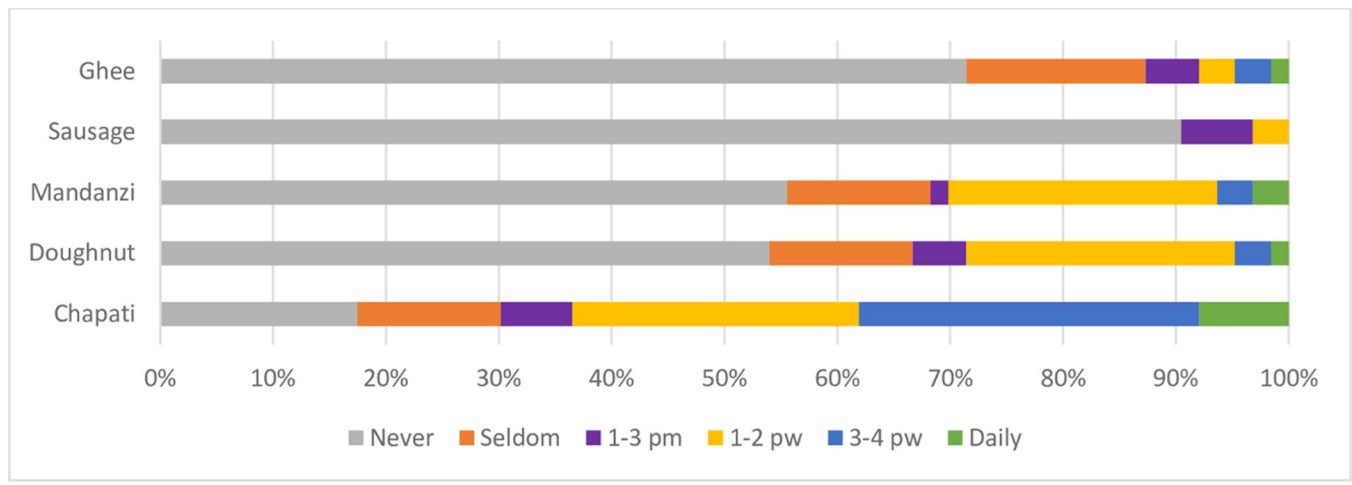

**Fig 4. Frequency of dietary intake of low nutrient density foods, such as oils and snacks (fats).** pm = per month; pw = per week.

anthropometric parameters such as abdominal fat composition or WHR, probably because since majority of the participants were within the normal range (Table 7). However, the only significant correlations observed were between the level of education and nutritional status, especially at lower levels, which is due to the fact that majority of our study participants had attained only either primary or secondary education, but with normal nutritional status. This is interestingly contradictory because, lower levels of education have been associated with having abnormal nutritional status and other disorders in other countries [40–43]. Our findings generally, seems to suggest that majority of our study participants still had normal body composition basing on their total body fat, muscle mass, bone mass and total body water analyses (Tables 3 and 4). This may also imply good management of the CKD condition and that the duration for which the patients have been under treatment is not so long. This was a bit unexpected, since CKD is usually associated with changes in body composition due to protein-energy wasting and cachexia, especially in advanced disease situation [44–46].

## Conclusion

Hemodialysis patients at Kiruddu National Referral Hospital, have altered nutrition status as evidenced by altered body weight for some patients and deranged micronutrient levels (K, Vitamin D, Phosphorus) arising from reduced dietary intake of major macronutrients (proteins) and micronutrients (vitamins). The body composition seems to be relatively normal for most of the patients. We recommend that hemodialysis patients should be regularly assessed for nutrition status, appropriately treated and educated about their nutritional status.

## Limitations

The study was not able to assess for dietary intake using the food records/dairies which is recommended by KODQ for assessment of dietary intake in CKD patients. This is because they are not feasible in illiterates and the patient can forget to record down what they ate. Also, the study was not able to determine the serum levels of the other proposed vitamins (Thiamine, pyridoxine, riboflavin, vitamin C and K) that are commonly deficient among CKD patients due to limited funding.

## Supporting information

**S1 Data.**
(XLSX)

## Acknowledgments

We are grateful for the assistance rendered to our study by the clinical staff of Kiruddu National Referral Hospital-Dialysis unit who assisted us in data collection and the technical team at Mulago clinical chemistry laboratory who analyzed our study samples. We also appreciate and thank the participants who unreservedly consented and agreed to be involved in our study.

## Author Contributions

**Conceptualization:** Fred Lawrence Sembajwe, Agnes Namaganda, Joshua Nfambi, Haruna Muwonge, Godfrey Katamba, Ritah Nakato, Prossy Nabachenje, Enid Kawala Kagoya, Annet Namubamba, Daniel Kiggundu, Brian Bitek, Robert Kalyesubula, Jehu Iputo.

**Data curation:** Fred Lawrence Sembajwe, Agnes Namaganda, Joshua Nfambi, Godfrey Katamba, Prossy Nabachenje, Brian Bitek.

**Formal analysis:** Fred Lawrence Sembajwe, Agnes Namaganda, Joshua Nfambi, Godfrey Katamba, Prossy Nabachenje, Brian Bitek.

**Funding acquisition:** Agnes Namaganda.

**Investigation:** Fred Lawrence Sembajwe, Agnes Namaganda, Joshua Nfambi, Haruna Muwonge, Godfrey Katamba, Ritah Nakato, Prossy Nabachenje, Enid Kawala Kagoya, Annet Namubamba, Daniel Kiggundu, Brian Bitek, Robert Kalyesubula.

**Methodology:** Fred Lawrence Sembajwe, Agnes Namaganda, Haruna Muwonge, Godfrey Katamba, Prossy Nabachenje, Daniel Kiggundu, Robert Kalyesubula.

**Project administration:** Agnes Namaganda.

**Resources:** Agnes Namaganda.

**Supervision:** Haruna Muwonge, Daniel Kiggundu, Robert Kalyesubula, Jehu Iputo.

**Validation:** Haruna Muwonge, Daniel Kiggundu, Robert Kalyesubula, Jehu Iputo.

**Visualization:** Agnes Namaganda, Robert Kalyesubula, Jehu Iputo.

**Writing – original draft:** Fred Lawrence Sembajwe, Agnes Namaganda.

**Writing – review & editing:** Fred Lawrence Sembajwe, Agnes Namaganda, Joshua Nfambi, Haruna Muwonge, Godfrey Katamba, Ritah Nakato, Prossy Nabachenje, Enid Kawala Kagoya, Annet Namubamba, Daniel Kiggundu, Brian Bitek, Robert Kalyesubula, Jehu Iputo.

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
