## [Decision Letter · Decision Letter 0]

23 Jun 2023

PONE-D-23-06538Dietary intake, body composition and micronutrient profile of patients on maintenance hemodialysis attending Kiruddu National Referral Hospital, Uganda. A cross sectional studyPLOS ONE

Dear Dr. Namaganda,

Thank you for submitting your manuscript to PLOS ONE. After careful consideration, we feel that it has merit but does not fully meet PLOS ONE’s publication criteria as it currently stands. Therefore, we invite you to submit a revised version of the manuscript that addresses the points raised during the review process.

ACADEMIC EDITOR: The current study assessed the dietary status of CKD patients on maintenance haemodialysis. the current study meets research ethical criteria, and experimental integrity. research question should not be bulleted but stated in sentence form as well as specific objectives.what inform the sample size of 63 in an institution which receives 120 CKD  patients monthly.what is the prevalence of CKD in study area.

Table 1 what is the difference between tertiary and university.

the result presentation, discussion, and conclusion should be based on the three main specific objectives.the study design of cross sectional makes it difficult to make emphatic conclusions.==============================

We look forward to receiving your revised manuscript.

Kind regards,

Samuel Asamoah Sakyi, Ph.D

Academic Editor

PLOS ONE

Journal Requirements:

“We are grateful for the funding that enabled this research work, received from the Fogarty International Center of the National Institutes of Health (NIH), U.S. Department of State’s Office of the U.S. Global AIDS Coordinator and Health Diplomacy (S/GAC), and President’s Emergency Plan for AIDS Relief (PEPFAR) under award number: IR25TWO11213”

“NA received funding from the Fogarty International Center of the National Institutes of Health (NIH), U.S. Department of State’s Office of the U.S. Global AIDS Coordinator and Health Diplomacy (S/GAC), and President’s Emergency Plan for AIDS Relief (PEPFAR) under award number: IR25TWO11213. The funders had no role in study design, data collection and analysis, decision to publish, or preparation of the manuscript.”

Reviewers'  comments:

Reviewer's Responses to Questions

**Comments to the Author**

1. Is the manuscript technically sound, and do the data support the conclusions?

Reviewer #1: Yes

Reviewer #2: Yes

2. Has the statistical analysis been performed appropriately and rigorously? 

Reviewer #1: Yes

Reviewer #2: Yes

3. Have the authors made all data underlying the findings in their manuscript fully available?

Reviewer #1: Yes

Reviewer #2: Yes

4. Is the manuscript presented in an intelligible fashion and written in standard English?

Reviewer #1: Yes

Reviewer #2: No

5. Review Comments to the Author

Reviewer #1: The authors presented clear and understandable information on the need to assess dietary intake, body composition and micronutrient profile of patients undergoing maintenance hemodialysis.

While the manuscript is well written, some issues are present:

1. The title in line 2 page 1 has a full stop after the country name and before “A cross sectional”. This is not appropriate for a title and should be replaced with a colon.

2. Abstract: The authors in line 29 page 2, line 50 page 3 noted “nutrition status” while in the introduction line 69 page 4 used “nutritional status” to connote the same thing. I suggest they change the “nutrition status” to “nutritional status” for uniformity in the entire manuscipt.

3. Line 37 - 38 “Micronutrient profile 38 assessment was measured with the COBAS Auto analyzer based on patients’ serum samples”. This sentence should be rephrased for clarity.

4. Introduction: The authors in line 57 to line 59 page 3 stated the prevalence of CKD regionally in Uganda but did not reference the prevalence of for eastern Uganda. It would be much appreciated if the reference for this prevalence is stated just like the other two.

5. Line 96 - 97 page 5. The authors spoke of numerous studies in relation to body composition determination but cited only a single paper. The sentence should be rephrased to connote that the information is only from a single source or additional references should be cited.

6. Line 161 page 8. The authors said, “…waist and circumference were obtained using a flexible measuring tape”. This sentence does not make sense and thus should be rephrased with the insertion of the required words to communicate the intended information.

7. Line 39-40 and 193. “T-test was used to make comparisons and 40 logistic regression to analyze the correlations”: The second part of this sentence seems a little ambiguous and needs to be clarified by the authors and the sentence rephrased.

8. I noted that there are missing data in some of the data, meanwhile the authors made no comment on this in the write up. For example; the total frequency for occupation in Table 1, and Abdominal fat level in Table 4 is 60, body mass index (BMI) is 61, and that of all micronutrients in Table 9 with the exception of Cl is 60. Let the authors please clarify this.

9. The authors are requested to take a second look at the discussion session and correct the grammatical mistakes.

10. The authors in line 50 - 51 page 3 and line 341-342 page 18 in the recommendation stated “. We recommend that hemodialysis patients should be regularly assessed for nutrition status, appropriately treated and health educated about their nutrition status” this sentence should be rephrased with the removal of “health” from before the “educated” and the replacement of “nutrition status” with nutritional status.

Reviewer #2: This is an interesting study regarding the dietary intake and body composition of patient with kidney diseases. Proper dietary balance is crucial for kidney patient as this may help in regulating body composition and keep kidney functioning.

I propose to address the following issues.

Line 106: Research questions and research objectives does not need to be written in bullet points. These bullet points need to be concise and written as wording.

Methodology:

Line 147: How the sample size was calculated need to be written in details in this section.

Line 157: The FFQ, that has been used, is validated in Uganda?

Line 164: The principle of body composition analyser needs to write in details so that the reader can understand this.

Line 170: How long the serum was stored, if for longer period, whether -80C was maintained or not. If not how the quality of the sample was ensured.

Results:

Line 240: In Table 3, unit of few variables like blood pressure, MUAC and others need to be included.

Line 265: "Born mass" need to be corrected.

Line 269: In table 8 unit of micronutrient concentration need to be included.

Discussion:

The discussion is written in right direction.

General Comments:

The research was done to understand about the dietary intake, body composition and micronutrient imbalance. However, it is arguable to me that, whether this research has added new insight or knowledge to the scientific world, as it is well known that CKD patient frequently face this kind of problem. It was better to design a study to resolve these complications that CKD patient frequently confronted with.

6. PLOS authors have the option to publish the peer review history of their article (what does this mean?). If published, this will include your full peer review and any attached files.

Reviewer #1: No

Reviewer #2: **Yes: **Md Kamrzzaman

---

## [Author Response · Author response to Decision Letter 0]

18 Jul 2023

ACADEMIC EDITOR Comments: The current study assessed the dietary status of CKD patients on maintenance haemodialysis. the current study meets research ethical criteria, and experimental integrity. 

1. Research question should not be bulleted but stated in sentence form as well as specific objectives.

Response: This has been addressed and both research objectives or questions are implied in the statement in lines between 108 and 110.

2. what informed the sample size of 63 in an institution which receives 120 CKD patients monthly. 

Response: It was a pilot study which is meant to pave way for a larger survey. This has been changed in the manuscript 

3. What is the prevalence of CKD in study area?

Response: The prevalence of CKD in the study area is 2.5% as shown in line 61 of the manuscript.

4. Table 1 what is the difference between tertiary and university.

Response: This has been clarified by adding diploma certificate in brackets for tertiary education as opposed to university education where a degree is usually attained.

5. the result presentation, discussion, and conclusion should be based on the three main specific objectives.

Response: Yes, the results and discussion presentation are based on the main of objectives of the study to find out about dietary intake, micronutrient profile and body composition of CKD patients

6. the study design of cross sectional makes it difficult to make emphatic conclusions.

Response: Yes, we have tried to make a modest conclusion since our pilot cross-sectional study design does not allow us to make very strong and confirmatory conclusions.

Reviewers' comments:

 Reviewer #1: The authors presented clear and understandable information on the need to assess dietary intake, body composition and micronutrient profile of patients undergoing maintenance hemodialysis.

While the manuscript is well written, some issues are present:

1. The title in line 2 page 1 has a full stop after the country name and before “A cross sectional”. This is not appropriate for a title and should be replaced with a colon.

Response: This has been corrected as suggested by the reviewer.

2. Abstract: The authors in line 29 page 2, line 50 page 3 noted “nutrition status” while in the introduction line 69 page 4 used “nutritional status” to connote the same thing. I suggest they change the “nutrition status” to “nutritional status” for uniformity in the entire manuscript.

Response: This has been corrected as suggested by the reviewer.

3. Line 37 - 38 “Micronutrient profile 38 assessment was measured with the COBAS Auto analyzer based on patients’ serum samples”. This sentence should be rephrased for clarity.

Response: This has been rephrased like this: “Serum micronutrient profile assessment was done using the COBAS Auto analyzer”.

4. Introduction: The authors in line 57 to line 59 page 3 stated the prevalence of CKD regionally in Uganda but did not reference the prevalence of for eastern Uganda. It would be much appreciated if the reference for this prevalence is stated just like the other two.

Response: The prevalence of Eastern Uganda was stated as being 12.5%; however, there was no prevalence stated for the Northern region of the country, because not many CKD-studies have been done in that part of the country. However, one study done among HIV patients found a prevalence of renal impairment of about 14.4% as indicated in line 61. 

5. Line 96 - 97 page 5. The authors spoke of numerous studies in relation to body composition determination but cited only a single paper. The sentence should be rephrased to connote that the information is only from a single source or additional references should be cited.

Response: This sentence has been rephrased and more references added as shown in lines 100 and 101.

6. Line 161 page 8. The authors said, “…waist and circumference were obtained using a flexible measuring tape”. This sentence does not make sense and thus should be rephrased with the insertion of the required words to communicate the intended information.

Response: This has been addressed and the statement has been rephrased as: “body weight, height and circumference were measured to the nearest 0.1Kg or 0.1cm respectively, using a portable weight scale (for weight); waist and mid-upper arm circumferences were obtained using a flexible measuring tape”.

7. Line 39-40 and 193. “T-test was used to make comparisons and 40 logistic regression to analyze the correlations”: The second part of this sentence seems a little ambiguous and needs to be clarified by the authors and the sentence rephrased.

Response: This sentence has been rephrased as shown in lines: 40-41 and 205-206.

8. I noted that there are missing data in some of the data, meanwhile the authors made no comment on this in the write up. For example; the total frequency for occupation in Table 1, and Abdominal fat level in Table 4 is 60, body mass index (BMI) is 61, and that of all micronutrients in Table 9 with the exception of Cl is 60. Let the authors please clarify this.

Response: The sample size was 63, however there was missing data for some parameters that is why some tables have a reduced number. For the tables where data was missing, analysis was adjusted to leave out the missing data. Missing data was not part of the analysis.

This comment of missing data has been included in the write-up. Line 216 and line 294

9. The authors are requested to take a second look at the discussion session and correct the grammatical mistakes.

Response: The typing errors in the discussion have been corrected and more statements have been added for more clarity.

10. The authors in line 50 - 51 page 3 and line 341-342 page 18 in the recommendation stated “. We recommend that hemodialysis patients should be regularly assessed for nutrition status, appropriately treated and health educated about their nutrition status” this sentence should be rephrased with the removal of “health” from before the “educated” and the replacement of “nutrition status” with nutritional status.

Response: This has been corrected 

Reviewer #2: This is an interesting study regarding the dietary intake and body composition of patient with kidney diseases. Proper dietary balance is crucial for kidney patient as this may help in regulating body composition and keep kidney functioning.

I propose to address the following issues.

Line 106: Research questions and research objectives does not need to be written in bullet points. These bullet points need to be concise and written as wording.

Response: This has been addressed and both research objectives or questions are implied in the statement in lines between 108 and 110.

Methodology:

Line 147: How the sample size was calculated need to be written in details in this section.

Response I did a pilot study. Have changed it in the manuscript

Line 157: The FFQ, that has been used, is validated in Uganda?

Response: The FFQ used in this study was adapted from the WHO FFQ and adapted for this study. It was checked for content, construct and internal validity. It was also tested and retested for consistency and inter rater reliability. This has been addressed in the manuscript with track changes at line 164-167

Line 164: The principle of body composition analyser needs to write in details so that the reader can understand this.

Response: This has been addressed in the manuscript as: It uses the principle of different impedances of the conductive and insulated parts of the body, calculates the weight and proportion of various components in the body, automatically analyzes the test data, and uses the chart method to visually illustrate physical health. Line 173-175 (manuscript with track changes).

Line 170: How long the serum was stored, if for longer period, whether -80C was maintained or not. If not how the quality of the sample was ensured.

Response: The serum was kept for a few hours at 4 degrees Celsius in the KNRH refrigerator before being transferred to the clinical chemistry laboratory at Mulago hospital for further analysis, and throughout the transportation, a cold chain environment was maintained by transporting the samples in cooled boxes with ice packs.

Results:

Line 240: In Table 3, unit of few variables like blood pressure, MUAC and others need to be included.

Response: The necessary units have been added including mmHg for blood pressure, cm for MUAC.

Line 265: "Born mass" need to be corrected.

Response: This has been corrected to Bone mass

Line 269: In table 8 unit of micronutrient concentration need to be included.

Response: The units have been added in the table as: ng/ml = nanograms/milliliters for vitamin D; g/dL = grams/liter for albumin; mEq/L = milliequivalents/Liter for electrolytes 

Discussion:

The discussion is written in right direction.

General Comments:

The research was done to understand about the dietary intake, body composition and micronutrient imbalance. However, it is arguable to me that, whether this research has added new insight or knowledge to the scientific world, as it is well known that CKD patient frequently face this kind of problem. It was better to design a study to resolve these complications that CKD patient frequently confronted with.

Response: Due to the unique economic situation in a third world country like Uganda, it was not so obvious to know exactly how CKD was impacting the nutritional status of people who might not be able to afford a balanced diet for instance, or people who might not be able to afford some of the recommended alternatives to the restricted foods for patients with CKD. This is why such studies are necessary to see how people in such situations are affected before interventional studies can be undertaken.

Kindly edit the funding statement to; 

“We received funding from the Fogarty International Center of the National Institutes of

Health (NIH), U.S. Department of State’s Office of the U.S. Global AIDS Coordinator and

Health Diplomacy (S/GAC), and President’s Emergency Plan for AIDS Relief (PEPFAR)

under award number: IR25TWO11213. The funders had no role in study design, data

collection and analysis, decision to publish, or preparation of the manuscript.”

---

## [Decision Letter · Decision Letter 1]

7 Sep 2023

Dietary intake, body composition and micronutrient profile of patients on maintenance hemodialysis attending Kiruddu National Referral Hospital, Uganda. A pilot study

PONE-D-23-06538R1

Dear Agnes Namaganda

We’re pleased to inform you that your manuscript has been judged scientifically suitable for publication and will be formally accepted for publication once it meets all outstanding technical requirements.

Kind regards,

Samuel Asamoah Sakyi, Ph.D

Academic Editor

PLOS ONE

Additional Editor Comments (optional):

Reviewers' comments:

Reviewer's Responses to Questions

**Comments to the Author**

1. authors have adequately addressed your comments raised in a previous round of review and I feel that this manuscript is now acceptable for publication, 

Reviewer #1: All comments have been addressed

Reviewer #2: (No Response)

2. Is the manuscript technically sound, and do the data support the conclusions?

Reviewer #1: Yes

Reviewer #2: Yes

3. Has the statistical analysis been performed appropriately and rigorously? 

Reviewer #1: Yes

Reviewer #2: Yes

4. Have the authors made all data underlying the findings in their manuscript fully available?

Reviewer #1: Yes

Reviewer #2: Yes

5. Is the manuscript presented in an intelligible fashion and written in standard English?

Reviewer #1: Yes

Reviewer #2: Yes

6. Review Comments to the Author

Reviewer #1: The authors presented clear and understandable information on the need to assess dietary intake, body composition and micronutrient profile of patients undergoing maintenance hemodialysis.

Requested revisions have all been adequately addressed by the authors.

Reviewer #2: The following issues need to be addressed.

1. Sample size calculation was not addressed in the manuscript. This need to be addressed.

2. The FFQ, that has been used, is validated in Uganda? This issue needs to be addressed.

3. How long the serum was stored, if for longer period, whether -80C was maintained or not. If not how the quality of the sample was ensured. This issue needs to be addressed.

7. PLOS authors have the option to publish the peer review history of their article (what does this mean?). If published, this will include your full peer review and any attached files.

Reviewer #1: No

Reviewer #2: **Yes: **Md Kamruzzaman

---

## [Editor Report · Acceptance letter]

11 Oct 2023

PONE-D-23-06538R1 

Dietary intake, body composition and micronutrient profile of patients on maintenance hemodialysis attending Kiruddu National Referral Hospital in Uganda: A cross sectional study 

Dear Dr. Namaganda:

I'm pleased to inform you that your manuscript has been deemed suitable for publication in PLOS ONE. Congratulations! Your manuscript is now with our production department. 

Kind regards, 

on behalf of

Dr. Samuel Asamoah Sakyi 

Academic Editor

PLOS ONE